# Adaptive Stochastic Optimization: From Sets to Paths

**Zhan Wei Lim**     **David Hsu**     **Wee Sun Lee**

Department of Computer Science, National University of Singapore
{limzhanw,dyhsu,leews}@comp.nus.edu.sg

## Abstract

Adaptive stochastic optimization (ASO) optimizes an objective function adaptively under uncertainty. It plays a crucial role in planning and learning under uncertainty, but is, unfortunately, computationally intractable in general. This paper introduces two conditions on the objective function, the *marginal likelihood rate bound* and the *marginal likelihood bound*, which, together with pointwise submodularity, enable efficient approximate solution of ASO. Several interesting classes of functions satisfy these conditions naturally, *e.g.*, the version space reduction function for hypothesis learning. We describe *Recursive Adaptive Coverage*, a new ASO algorithm that exploits these conditions, and apply the algorithm to two robot planning tasks under uncertainty. In contrast to the earlier submodular optimization approach, our algorithm applies to ASO over both sets and *paths*.

## 1  Introduction

A hallmark of an intelligent agent is to learn new information as the world unfolds and to improvise by fusing the new information with prior knowledge. Consider an autonomous unmanned aerial vehicle (UAV) searching for a victim lost in a jungle. The UAV acquires new information on the victim's location by scanning the environment with noisy onboard sensors. How can the UAV plan and adapt its search strategy in order to find the victim as fast as possible? This is an example of *stochastic optimization*, in which an agent chooses a sequence of actions under uncertainty in order to optimize an objective function. In *adaptive* stochastic optimization (ASO), the agent's action choices are conditioned on the outcomes of earlier choices. ASO plays a crucial role in planning and learning under uncertainty, but it is, unfortunately, computationally intractable in general [5].

Adaptive submodular optimization provides a powerful tool for approximate solution of ASO and has several important applications, such as sensor placement, active learning, *etc.* [5]. However, it has been so far restricted to optimization over a set domain: the agent chooses a subset out of a finite set of items. This is inadequate for the UAV search, as the agent's consecutive choices are constrained to form a *path*. Our work applies to ASO over both sets and paths.

Our work aims to identify subclasses of ASO and provide conditions that enable efficient near-optimal solution. We introduce two conditions on the objective function, the *marginal likelihood rate bound* (MLRB) and the *marginal likelihood bound* (MLB). They enable efficient approximation of ASO with pointwise submodular objective functions, functions that satisfy a "diminishing return" property. MLRB is different from adaptive submodularity; we prove that adaptive submodularity does not imply MLRB and *vice versa*. While there exist functions that do not satisfy either the adaptive submodular or the MLRB condition, all pointwise submodular functions satisfy the MLB condition, albeit with different constants.

We propose *Recursive Adaptive Coverage* (RAC), a polynomial-time approximation algorithm that guarantees near-optimal solution of ASO over either a set or a path domain, if the objective function satisfies the MLRB or the MLB condition and is pointwise monotone submodular. Since MLRB differs from adaptive submodularity, the new algorithm expands the set of problems that admit efficient approximate solutions, even for ASO over a set domain. We have evaluated RAC in simulation on two robot planning tasks under uncertainty and show that RAC performs well against several commonly used heuristic algorithms, including greedy algorithms that optimize information gain.

## 2 Related Work

Submodular set function optimization encompasses many hard combinatorial optimization problems in operation research and decision making. Submodularity implies a diminishing return effect where adding an item to a smaller set is more beneficial than adding the same item to a bigger set. For example, adding a new temperature sensor when there are few sensors helps more in mapping temperature in a building than when there are already many sensors. Submodular functions can be efficiently approximated using a greedy heuristic [11]. Recent works have incorporated stochasticity to submodular optimization [1, 5] and generalized the problem from sets optimization to path optimization [2].

Our work builds on progress in submodular optimization on paths to solve the adaptive stochastic optimization problem on paths. Our RAC algorithm share a similar structure and analysis as the RAId algorithm in [10] that is used to solve adaptive informative path planning (IPP) problems without noise. In fact, noiseless adaptive IPP is a special case of adaptive stochastic optimization problems on paths that satisfies the marginal likelihood rate bound condition. We can derive the same approximation bound using the results in Section 6. Both works are inspired by the algorithm in [8] used to solve the Adaptive Traveling Salesperson (ATSP) problem. In the ATSP problem, a salesperson has to service a subset of locations with demand that is not known in advance. However, the salesperson knows the prior probabilities of the demand at each location (possibly correlated) and the goal is to find an adaptive policy to service all locations with demand.

Adaptive submodularity [5] generalizes submodularity to stochastic settings and gives logarithmic approximation bounds using a greedy heuristic. It was also shown that no polynomial time algorithm can compute approximate solution of adaptive stochastic optimization problems within a factor of $O(|X|^{1-\epsilon})$ unless $PH = \sum_2^p$, that is the polynomial-time hierarchy collapses to its second level [5]. Many Bayesian active learning problems can be modeled by suitable adaptive submodular objective functions [6, 4, 3]. However, [3] recently proposed a new stochastic set function for active learning with a general loss function that is not adaptive monotone submodular. This new objective function satisfies the marginal likelihood bound with nontrivial constant $G$.

Adaptive stochastic optimization is a special case of the Partially Observable Markov Decision Process (POMDP), a mathematical principled framework for reasoning under uncertainty [9]. Despite recent tremendous progress in offline [12] and online solvers [14, 13], most partially observable planning problems remain hard to solve.

## 3 Preliminaries

We now describe the adaptive stochastic optimization problem and use the UAV search and rescue task to illustrate our definitions. Let $X$ be the set of actions and let $O$ be the set of observations. The agent operates in a world whose events are determined by a static state called the *scenario*, denoted as $\phi : X \to O$. When the agent takes an action $x \in X$, it receives an observation $o = \phi(x) \in O$ that is determined by an initially unknown scenario $\phi$. We denote a random scenario as $\Phi$ and use a prior distribution $p(\phi) := \mathbb{P}[\Phi = \phi]$ over the scenarios to represent our prior knowledge of the world.

For *e.g.*, in the UAV task, the actions are flying to various locations, observations are the possible sensors' readings, and a scenario is a victim's position. When the UAV flies to a particular location $x$, it observes its sensors' readings $o$ that depends on actual victim's position $\phi$. Prior knowledge about the victim's position can be encoded as a probability distribution over the possible victim's positions.

After taking actions $x_1, x_2, \ldots$ and receiving observations $o_1, o_2, \ldots$ after each action, the agent has a *history* $\psi = \{(x_1, o_1), (x_2, o_2), \ldots \}$. We say that a scenario $\phi$ is consistent with a history $\psi$ when the actions and corresponding observations of the history never contradict with the $\phi$, *i.e.* $\phi(x) = o$ for all $(x, o) \in \psi$. We denote this by $\phi \sim \psi$. We can also say that a history $\psi'$ is consistent with another history $\psi$ if $\text{dom}(\psi') \supset \text{dom}(\psi)$ and $\psi'(x) = \psi(x)$ for all $x \in \text{dom}(\psi)$, where $\text{dom}(\psi)$ is the set of actions taken in $\psi$. For example, a victim's position $\phi$ has not been ruled out given the sensors readings at various locations $\psi$ when $\phi \sim \psi$.

An agent's goal can be characterized by a stochastic set function $f : 2^X \times O^X \to \mathbb{R}$, which measures progress toward the goal given the actions taken and the true scenario. In this paper, we assume that $f$ is pointwise monotone on finite domain. *i.e.*, $f(A, \phi) \le f(B, \phi)$ for any $\phi$ and for

all $A \subseteq B \subseteq X$. An agent achieves its goal and *covers* $f$ when $f$ has maximum value after taking actions $S \subseteq X$ and given it is in scenario $\phi$, *i.e.*, $f(S, \phi) = f(X, \phi)$. For example, the objective function can be the sum of prior probabilities of impossible victim's positions given a history. The UAV finds the victim when all except the true victim's position are impossible.

An agent's strategy for adaptively taking actions is a policy $\pi$ that maps a history to its next action. A policy terminates when there is no next action to take for a given history. We say that a policy $\pi$ *covers* the function $f$ when the agent executing $\pi$ always achieves its goal upon termination. That is, $f(\text{dom}(\psi), \phi) = f(X, \phi)$ for all scenarios $\phi \sim \psi$, where $\psi$ is the history when the agent executes $\pi$. For example, a policy $\pi$ tells the UAV where to fly to next given the locations visited and whether it has a positive sensor at those locations or not and it covers the objective function when the UAV executing it always find the victim.

Formally, an adaptive stochastic optimization problem *on paths* consists of the tuple $(X, d, p, O, r, f)$, the set of actions $X$ is the set of locations the agent can visit, $r$ is the starting location of the agent, and $d$ is a metric that gives the distance between any pair of locations $x, x' \in X$. The cost of the policy $\pi$, $C(\pi, \phi)$, is the length of the path starting from location $r$ traversed by the agent until the policy terminates, when presented with scenario $\phi$, *e.g.*, the distance traveled by UAV executing policy $\pi$ for a particular true victim position. We want to find a policy $\pi$ that minimizes the cost of traveling to cover the function. We formally state the problem:

**Problem 1.** *Given an adaptive stochastic optimization problem on paths $\mathcal{I} = (X, d, p, O, r, f)$, compute an adaptive policy that minimizes the expected cost*

$$C(\pi) = \text{E}[C(\pi, \phi)] = \sum_\phi C(\pi, \phi) p(\phi). \tag{1}$$

*subject to $f(dom(\psi), \phi') = f(X, \phi')$, where $\psi$ is the history encountered when executing $\pi$ on $\phi'$, for all $\phi$'.*

Adaptive stochastic optimization problems *on sets* can be formally defined by a tuple, $(X, c, p, O, f)$. The set of actions $X$ is a set of items that an agent may select. Instead of a distance metric, the cost of selecting an item is defined by a cost function $c : X \to \mathbb{R}$ and the cost of a policy $C(\pi, \phi) = \sum_{x \in S} c(x)$, where $S$ is the subset of items selected by $\pi$ when presented with scenario $\phi$.

## 4 Classes of Functions

This section introduces the classes of objective functions for adaptive stochastic optimization problems and gives the relationship between them.

Given a finite set $X$ and a function on subsets of $X$, $f : 2^X \to \mathbb{R}$, the function $f$ is submodular if $f(A) + f(B) \geq f(A \cup B) + f(A \cap B)$ for all $A, B \subseteq X$. Let $f(S, \phi)$ be a stochastic set function. If $f(S, \phi)$ is submodular for each fixed scenario $\phi \in O^X$, then $f$ is *pointwise submodular*.

Adaptive submodularity and monotonicity generalize submodularity and monotonicity to stochastic settings where we receive random observations after selecting each item [6]. We define the expected marginal value of an item $x$ given a history $\psi$, $\triangle(x|\psi)$ as: $\triangle(x|\psi) = \text{E}\left[ f(\text{dom}(\psi) \cup \{x\}, \Phi) - f(\text{dom}(\psi), \Phi) \mid \Phi \sim \psi \right]$. A function $f : 2^X \times O^X \to \mathbb{R}$ is *adaptive monotone* with respect to a prior distribution $p(\phi)$ if , for all $\psi$ such that $\mathbb{P}[\Phi \sim \psi] > 0$ and all $x \in X$, it holds that $\triangle(x|\psi) \geq 0$. *i.e.* the expected marginal value of any fixed item is nonnegative. Function $f$ is *adaptive submodular* with respect to a prior distribution $p(\phi)$ if, for all $\psi$ and $\psi'$ such that $\psi' \sim \psi$ and for all $x \in X \backslash \text{dom}(\psi')$, it holds that $\triangle(x|\psi) \geq \triangle(x|\psi')$. *i.e.* the expected marginal value of any fixed item does not increase as more items are selected. A function can be adaptive submodular with respect to a certain distribution $p$ but not be pointwise submodular. However, it must be pointwise submodular if it is adaptive submodular with respect to all distributions.

We denote $\hat{f}(S, \psi) = \min_{\phi \sim \psi} f(S, \phi)$ as the worst case value of $f$ given a history and $p(\psi) := \mathbb{P}[\Phi \sim \psi]$ as the marginal likelihood of a history. The *marginal likelihood rate bound* (MLRB) condition requires a function $f$ such that: For all $\psi' \sim \psi$, if $p(\psi') \leq 0.5 p(\psi)$ then ,

$$Q - \hat{f}(\text{dom}(\psi'), \psi) \leq \frac{1}{K}\left(Q - \hat{f}(\text{dom}(\psi), \psi)\right), \tag{2}$$

except for scenarios already covered, where $K > 1$ and $Q \geq \max_\phi f(X, \phi)$ is a constant upper bound for the maximum value of $f$ for all scenarios.

Intuitively, this condition means that the worst case remaining objective value decreases by a constant fraction whenever the marginal likelihood of history decreases by at least half.

**Example:** The version space reduction function $\mathcal{V}$ with arbitrary prior is adaptive submodular and monotone [5]. Furthermore, it satisfies the MLRB. The version space reduction function $\mathcal{V}$ is defined as:

$$\mathcal{V}(S, \phi) = 1 - \sum_{\phi' \sim \phi(S)} p(\phi') \tag{3}$$

for all scenario $\phi$, $S \subseteq X$ and $\phi(S)$ gives the history of visiting locations $x$ in $S$ when the scenario is $\phi$. The version space reduction function is often used for active learning, where the true hypothesis is identified once all the scenarios are covered. We present the proof that the version space reduction function satisfies the MLRB condition (and all other proofs) in the supplementary material.

**Proposition 1.** *The version space function $\mathcal{V}$ satisfies the MLRB with constants $Q = 1$ and $K = 2$.*

The following proposition teases apart the relationship between the MLRB condition and adaptive submodularity.

**Proposition 2.** *Adaptive submodularity does not imply the MLRB condition, and vice versa.*

The *marginal likelihood bound* (MLB) condition requires that there exists some constant $G$, such that for all $\psi$,

$$f(X, \phi) - \hat{f}(\mathrm{dom}(\psi), \psi) \leq G \cdot p(\psi). \tag{4}$$

In other words, the worst remaining objective value must be less than the marginal likelihood of its history multiplied by some constant $G$. Our quality of solution depends on the constant $G$. The smaller the constant $G$, the better the approximation bound.

We can make any adaptive stochastic optimization problem satisfy the MLB with a large enough constant $G$. To trivially ensure the bound of MLB, we set $G = Q \cdot 1/\delta$, where $\delta = \min_\phi p(\phi)$. Hence, $Q \leq G \cdot p(\psi)$ unless we have visited all locations and covered the function by definition.

**Example:** The version space reduction function $\mathcal{V}$ can be interpreted as the expected $0 - 1$ loss of a random scenario $\phi' \sim \psi$ differing from true scenario $\phi$. The loss is counted as one whenever $\phi' \neq \phi$. For example, a pair of scenarios that differ in observation at one location has the same loss of 1 as another pair that differs in all observations. Thus, it can be useful to assign different loss to different pair of scenarios with a general loss function. The generalized version space reduction function is defined as: $f_L(S, \phi) = \mathrm{E}_{\phi'}\left[L(\phi, \phi')\mathbf{1}(\phi(S) \neq \phi'(S))\right]$, where $\mathbf{1}(\cdot)$ is an indicator function and $L : O^X \times O^X \to \mathbb{R}_{\geq 0}$ is a general loss function that satisfies $L(\phi', \phi) = L(\phi, \phi')$ and $L(\phi, \phi') = 0$ if $\phi = \phi'$. The generalized version space reduction function is not adaptive submodular [3] and does not satisfy the MLRB condition. However, it satisfies condition MLB with a non-trivial constant $G$.

**Proposition 3.** *The generalized version space reduction function $f_L$ satisfies MLB with $G = \max_{\phi, \phi'} L(\phi, \phi')$.*

## 5 Algorithm

Adaptive planning is computationally hard due to the need to consider every possible observation after each action. RAC assumes that it always receive the most likely observation to simplify adaptive planning. RAC is a recursive algorithm that partially covers the function in each step and repeats on the residual function until the entire function is covered.

In each recursive step, RAC uses the mostly like observation assumption to transform adaptive stochastic optimization problem into a submodular orienteering problem to generate a tour and traverse it. If the assumption is true throughout the tour, then RAC achieves the required partial coverage. Otherwise, RAC receives some observation that has probability less than half (since only the most likely observation has probability at least half), the marginal likelihood of history decreases by at least half, and the MLRB and MLB conditions ensures that substantial progress is made towards covering the function.

Submodular orienteering takes a submodular function $g : X \to \mathbb{R}$ and a metric on $X$ and gives the minimum cost path $\tau$ that covers function $g$ such that $g(\tau) = g(X)$. We now describe the submodular orienteering problem used in each recursive step. Given the current history $\psi$, we construct a restricted set of location-observation pairs, $Z = \{(x, o) : (x, o) \notin$

$\psi, o$ is the most likely observation at $x$ given $\psi$}. Using ideas from [7], we construct a submodular function $g_\nu^* : 2^Z \to \mathbb{R}$ to be used in the submodular orienteering problem. Upon completion of the recursive step, we would like the function to be either covered or have value at least $\nu$ for all scenarios consistent with $\psi \cup Z'$ where $Z'$ is the selected subset of $Z$. We first restrict $\phi$ to a subset of scenarios $\Psi$ that are consistent with $\psi$. To simplify, we transform the function so that its maximum value for all $\phi$ is at least $\nu$ by defining $f_\nu(S, \phi) = f(S, \phi) + (\nu - f(X, \phi))$ whenever $f(X, \phi) < \nu$ and $f_\nu(S, \phi) = f(S, \phi)$ otherwise. For $Z' \subseteq Z$, we now define $g_\nu(Z', \phi) = f_\nu(\text{dom}(\psi \cup Z'), \phi)$ if $Z'$ is consistent with $\phi$ and $g_\nu(Z', \phi) = f_\nu(X, \phi)$ otherwise. Finally, we construct the submodular function $g_\nu^*(Z') = 1/|\Psi| \sum_{\phi \in \Psi} \min(\nu, g_\nu(Z', \phi))$. The constructions have the following properties that guarantees the effectiveness of the recursive steps of RAC.

**Proposition 4.** *Let $f$ be a pointwise monotone submodular function. Then $g_\nu$ is pointwise monotone submodular and $g_\nu^*$ is monotone submodular. In addition $g_\nu^*(Z') \geq \nu$ if and only if $f$ is either covered or have value at least $\nu$ for all scenarios consistent with $\psi \cup Z'$.*

We can replace $g_\nu^*$ by a simpler function if $f$ satisfies a *minimal dependency* property where the value of function $f$ depends only on the history, *i.e.* $f(\text{dom}(\psi), \phi') = f(\text{dom}(\psi), \phi)$ for all $\phi, \phi' \sim \psi$. We define a new submodular set function $g_\nu^m(Z') = g_\nu(Z', Z \cup \psi)$.

**Proposition 5.** *When $f$ satisfies minimal dependency, $g_\nu^m(Z') \geq \nu$ implies $g_\nu^*(Z') \geq \nu$.*

RAC needs to guard against committing to costly plan made under the most likely observation assumption which is bound to be wrong eventually. RAC uses two different mechanisms for hedging. For MLRB, instead of requiring complete coverage, we solve partial coverage using a submodular path optimization problem $g_{(1-1/K)Q}^*$ so that $f(S) \geq (1-1/K)Q$ for all consistent scenarios under the most likely observation assumption in each recursive step. For MLB, we solve submodular orienteering for complete coverage of $g_Q^*$ but also solve for the version space reduction function with 0.5 as the target, $\mathcal{V}_{0.5}^*$, as a hedge against over-commitment by the first tour when the function is not well aligned with the probability of observations. The cheaper tour is then traversed by RAC in each recursive step.

We define the informative observation set $\Omega_x$ for every location $x \in X$: $\Omega_x = \{ o \mid p(o|x) \leq 0.5 \}$. RAC traverses the tour and adaptively terminates when it encounters an informative observation. Subsequent recursive calls work on the residual function $f'$ and normalized prior $p'$. Let $\psi$ be the history encountered so far just before the recursive call, for any set $S \supset \text{dom}(\psi)$ $f'(S, \phi) = f(S, \phi) - f(\text{dom}(\psi), \phi)$. We assume that function $f$ is integer-valued. The recursive step is repeated until the residual value $Q' = 0$. We give the pseudocode of RAC in Algorithm 1. We give details of SUBMODULARPATH procedure and prove its approximation bound in supplementary material.

---

**Algorithm 1** RAC

**procedure** RECURSERAC$(p, f, Q)$
    **if** $\max_{\phi \in \{\phi'|p(\phi')>0\}} f(X, \phi) = 0$ **then**
        **return**
    $\tau \leftarrow$ GENTOUR$(p, f, Q)$
    $\psi \leftarrow$ EXECUTEPLAN$(\tau)$
    $p' \leftarrow \frac{p(\psi|\phi)p(\phi)}{p(\psi)}$
    $f' \leftarrow f(Y, \phi) - f(\tau, \phi)$
    $Q' \leftarrow Q - \min_\phi f(\tau, \phi)$ for all $\psi \sim \phi$
    RECURSERAC$(p', f', Q')$
**procedure** EXECUTEPLAN$(\tau)$
    **repeat**
        Visit next location $x$ in $\tau$ and observe $o$.
    **until** $o \in \Omega_x$ or end of tour.
    Move to location $x_t = r$.
    **return** history encountered $\psi$.

**procedure** GENTOUR$(p, f, Q)$
    **if** $f$ satisfies MLB **then**
        $\tau_f \leftarrow$ SUBMODULARPATH$(g_Q^*)$
        **if** $\max_\phi p(\phi) \leq 0.5$ **then**
            $\tau_{vs} \leftarrow$ SUBMODULARPATH$(\mathcal{V}_{0.5}^*)$
            $\tau \leftarrow \arg\min_{\tau_f, \tau_{vs}}(W(\tau'))$
        **else**
            $\tau \leftarrow \tau_f$
    **else**
        $\tau \leftarrow$ SUBMODULARPATH$(g_{(1-1/K)Q}^*)$
    **return** $\tau$ where $\tau = (x_0, x_1, \ldots, x_t)$ and $x_0 = x_t = r$

---

## 6  Analysis

We give the performance guarantees for applying RAC to adaptive stochastic optimization problem on paths that satisfy MLRB and MLB.

**Theorem 1.** *Assume that $f$ is an integer-valued pointwise submodular monotone function. If $f$ satisfies MLRB condition, then for any constant $\epsilon > 0$ and an instance of adaptive stochastic optimization problem on path optimizing $f$, RAC computes a policy $\pi$ in polynomial time such that*

$$C(\pi) = O((\log|X|)^{2+\epsilon} \log Q \log_K Q)C(\pi^*)),$$

*where $Q$ and $K > 1$ are constants that satisfies Equation* (2).

**Theorem 2.** *Assume that prior probability distribution $p$ is represented as non-negative integers with $\sum_\phi p(\phi) = P$ and $f$ is an integer-valued pointwise submodular monotone function. If $f$ satisfies MLB, then for any constant $\epsilon > 0$ and an instance of adaptive stochastic optimization problem on path optimizing $f$, RAC computes a policy $\pi$ for in polynomial time such that*

$$C(\pi) = O((\log|X|)^{2+\epsilon}(\log P + \log Q) \log G)C(\pi^*)),$$

*where $Q = \max_\phi f(X, \phi)$.*

For adaptive stochastic optimization problems on subsets, we achieve tighter approximation bounds by replacing the bound of submodular orienteering with greedy submodular set cover.

**Theorem 3.** *Assume $f$ is an integer-valued pointwise submodular and monotone function. If $f$ satisfies MLRB condition, then for an instance of adaptive stochastic optimization problem on subsets optimizing $f$, RAC computes a policy $\pi$ in polynomial time such that*

$$C(\pi) = 4(\ln Q + 1)(\log_K Q + 1)C(\pi^*),$$

*where $Q$ and $K > 1$ are constants that satisfies Equation* (2).

**Theorem 4.** *Assume $f$ is an integer-valued pointwise submodular and monotone function and $\delta = \min_\phi p(\phi)$. If $f$ satisfies MLB condition, then for an instance of adaptive stochastic optimization problem on subsets optimizing $f$, RAC computes a policy $\pi$ in polynomial time such that*

$$C(\pi) = 4(\ln 1/\delta + \ln Q + 2)(\log G + 1)C(\pi^*)),$$

*where $Q = \max_\phi f(X, \phi)$.*

## 7    Application: Noisy Informative Path Planning

In this section, we apply RAC to solve adaptive informative path planning (IPP) problems with noisy observations. We reduce an adaptive noisy IPP problem to an *Equivalence Class Determination* (ECD) problem [6] and apply RAC to solve it near-optimally using an objective function that satisfies MLRB condition. We evaluate this approach on two IPP tasks with noisy observations.

In an *informative path planning* (IPP) problem, an agent seeks a path to sense and gather information from its environment. An IPP problem is specified as a tuple $\mathcal{I} = (X, d, H, p_h, O, \mathcal{Z}_h, r)$, the definitions for $X, d, O, r$ are the same as adaptive stochastic optimization problem on path. In addition, there is a finite set of hypotheses, $H$, and a prior probability over them, $p(h)$. We also have a set of probabilistic observation functions $\mathcal{Z}_h = \{Z_x \mid x \in X\}$, with one observation function $Z_x(h, o) = p(o|x, h)$ for each location $x$. The goal of IPP problem is to identify the true hypothesis.

### 7.1    Equivalence Class Determination Problem

An *Equivalence Class Determination* (ECD) problem consists of a set of hypotheses $H$ and a set of equivalence classes $\{\mathcal{H}_1, \mathcal{H}_2, \ldots, \mathcal{H}_m\}$ that partitions $H$. Its goal is to identify which equivalence class the true hypothesis lies in by moving to locations and making observations with the minimum expected movement cost. ECD problem has been applied to noisy Bayesian active learning to achieve near-optimal performance. Noisy adaptive IPP problem can also be reduced to an ECD instance when it is always possible to identify the true hypothesis in IPP problem.

To differentiate between the equivalence classes, we use the Gibbs error objective function (called the edge-cutting function in [6]). The idea is consider the ambiguities between pairs of hypotheses in different equivalence class, and to visit locations and make observations to disambiguate between them. The set of pairs of hypotheses in different classes is $\mathcal{E} = \cup_{1 \le i < j \le m}\{\{h', h''\} : h' \in \mathcal{H}_i, h'' \in \mathcal{H}_j\}$. We disambiguate a pair $\{h', h''\}$ when we make an observation $o$ at a location $x$ and either $h'$ or $h''$ is inconsistent with the observation, $Z'_x(h', o) = 0$ or $Z'_x(h'', o) = 0$. The set of pairs disambiguated by visiting a location $x$ when hypothesis $h \in H'$ is true is given by

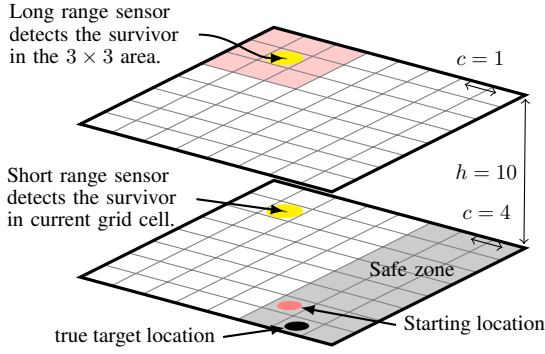

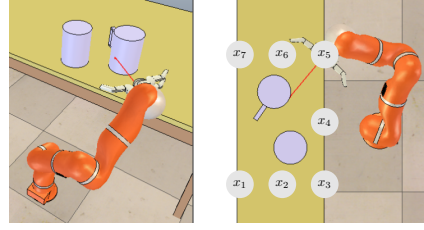

Figure 2: Grasp the cup with a handle top, the side view (left) and the top view (right).

Figure 1: UAV Search and Rescue

$\mathcal{E}_x(h) = \{\{h', h''\} : Z'_x(h, o) = 1, Z'_x(h', o) = 0 \text{ or } Z'_x(h'', o) = 0\}$. We define a weight function $w : \mathcal{E} \to \mathbb{R}_{\geq 0}$ as $w(\{h', h''\}) = p'(h') \cdot p'(h'')$. We can now define the Gibbs error objective function: $f_{GE}(\bar{Y}, h) = W(\cup_{x \in Y} \mathcal{E}_x(h))$, where $W(\mathcal{E}') = \sum_{e \in \mathcal{E}'} w(e)$, $Y$ is the set of location visited and $h \in H'$.

**Proposition 6.** *The Gibbs error function $f_{GE}$ is pointwise submodular and monotone. In addition, it satisfies condition MLRB with constants $Q = W(\mathcal{E}) = 1 - \sum_{i=1}^{m}(p(\mathcal{H}_i))^2$, the total weight of ambiguous pairs of hypotheses, and $K = 2$.*

The first step to reduce adaptive noisy IPP instance $\mathcal{I}$ to ECD instance $E$ is to create a noiseless IPP problem $\mathcal{I}' = (X, d, H', p', O, \mathcal{Z}', r)$ from a noisy IPP instance $\mathcal{I} = (X, d, H, p, O, \mathcal{Z}, r)$ is by creating a hypothesis for every possible observation vector. Each hypothesis $h' \in H'$ is an observation vector $h' = (o_1, o_2, \ldots, o_{|X|})$ and the new hypothesis space $H'$ is $H' = O^{|X|}$. Next, for each hypothesis $h_i \in H$, we create an equivalence class $\mathcal{H}_i = \left\{ (o_1, o_2, \ldots, o_{|X|}) \Big| \prod_{j=1}^{|X|} Z_{x_j}(h_i, o_j) > 0 \right\}$ that consists of all observation vectors $h' = (o_1, o_2, \ldots, o_{|X|}) \in H'$ that are possible with hypothesis $H_i$. When we can always identify the true underlying hypothesis $h \in H$, the equivalence classes is a partition on the set $H'$.

## 7.2 Experiments

We evaluate RAC in simulation on two noisy IPP tasks modified from [10]. We highlight the modifications and give the full description in the supplementary material. In a variant of the UAV search and rescue task (see Figure 1), there is a safe zone (marked grey in Figure 1) where the victim is deemed to be safe if we know that he is in it. otherwise we need to know the exact location of the victim. The equivalence classes task are the safe zone and every location outside of it. Furthermore, the long range sensor may report the wrong reading with probability of 0.03.

In a noisy variant of the grasping task, the laser range finder has a 0.85 chance of detecting the correct discretized value $x$, 0.05 chance of $\pm 1$ errors each, and 0.025 chance of $\pm 2$ errors each. The robot gripper is fairly robust to estimation error of the cup handle's orientation. For each cup, we partition the cup handle orientation into regions of 20 degrees each. We only need to know the region that contains cup handle. The equivalence classes here are the regions. However, it is not always possible to identify the true region due to observation noise. We can still reduce to ECD problem by associating each observation vector to its most likely equivalence class.

We now describe our baselines algorithms. Define information gain to be reduction in Shannon entropy of the equivalence classes, the *information gain* (IG) algorithm, greedily picks the location that maximizes the expected information gain, where the expectation is taken over all possible observations at the location. To account for movement cost, the *information gain* (IG-Cost) algorithm greedily picks the location that maximizes expected information gain per unit movement cost. Both IG and IG-Cost do not reason over the long term but achieve limited adaptivity by replanning in each step. The *Sampled*-RAId algorithm is as described in [10].

We evaluate IG, IG-Cost, *Sampled*-RAId, and RAC with version space reduction (RAC-$\mathcal{V}$) and Gibbs error (RAC-$GE$) objectives. RAC-$GE$ has theoretical performance guarantees for the noisy adaptive

IPP problem. Under the MLRB condition, RAC-$\mathcal{V}$ can also be shown to have a similar performance bound. However RAC-$GE$ optimizes the target function directly and we expect that optimizing the target function directly would usually have better performance in practice. Even though the version space reduction function and Gibbs error function are adaptive submodular, the greedy policy in [5] is not applicable as the movement cost per step depends on the paths and is not fixed. If we ignore movement cost, a greedy policy on the version space reduction function is equivalent to generalized binary search, which is equivalent to IG [15] for the UAV task where the prior is uniform and there are two observations.

We set all algorithms to terminate when the Gibbs error of the equivalence classes is less than $\eta = 10^{-5}$. The Gibbs error corresponds to the exponentiated Rényi entropy (order 2) and also the prediction error of a Gibbs classifier that predicts by sampling a hypothesis from the prior. We run 1000 trials with the true hypothesis sampled randomly from the prior for the UAV search task and 3000 trials for the grasping task as its variance is higher. For *Sampled*-RAId, we set the number of samples to be three times the number of hypothesis.

For performance comparison, we pick 15 different thresholds $\gamma$ (starting from $1 \times 10^{-5}$ and doubling $\gamma$ each step) for Gibbs error of the equivalence classes and compute the average cost incurred by each algorithm to reduce Gibbs error to below each threshold level $\gamma$. We plot the average cost with 95% confidence interval for the two IPP tasks in Figures 3 and 4. For the grasping task, we omit trials where the minimum Gibbs error possible is greater than $\gamma$ when we compute the average cost for that specific $\gamma$ value. For readability, we omit results due to IG from the plots when it is worse than other algorithms by a large margin, which is all of IG in the grasping task. From our experiments, RAC-$GE$ has the lowest average cost for both tasks at almost every $\gamma$. The RAC-$\mathcal{V}$ has very close results while the other algorithms, *Sampled*-RAId, IG-Cost and IG do not perform as well for both the UAV search and grasping task.

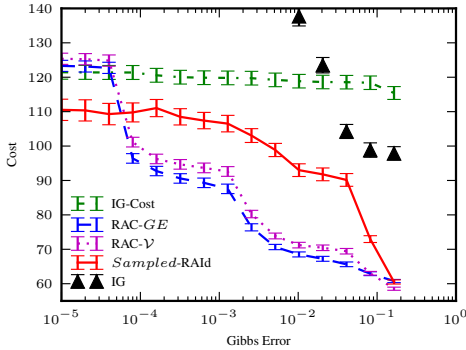

Figure 3: UAV search and rescue

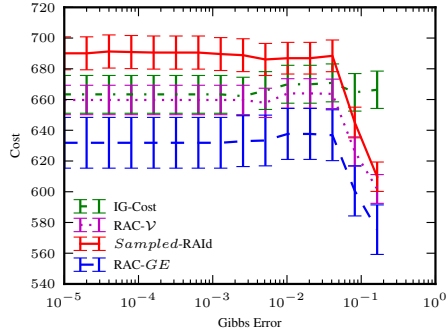

Figure 4: Grasping

## 8 Conclusion

We study approximation algorithms for adaptive stochastic optimization over both sets and paths. We give two conditions on pointwise monotone submodular functions that are useful for understanding the performance of approximation algorithms on these problems: the MLB and the MLRB. Our algorithm, RAC, runs in polynomial time with an approximation ratio that depends on the constants characterizing these two conditions. The results extend known results for adaptive stochastic optimization problems on sets to paths, and enlarges the class of functions known to be efficiently approximable for both problems. We apply the algorithm to two adaptive informative path planning applications with promising results.

**Acknowledgement** This work is supported in part by NUS AcRF grant R-252-000-587-112, National Research Foundation Singapore through the SMART Phase 2 Pilot Program (Subaward Agreement No. 09), and US Air Force Research Laboratory under agreement number FA2386-15-1-4010.

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
