[Supplementary Material]

# Supplementary Material

## 0.1 Proofs for examples of adaptive stochastic optimization problem

**Proposition 1.** *The version space function $\mathcal{V}$ satisfies MLRB with constants $Q = 1$ and $K = 2$*

*Proof.* We need to show

$$Q - \min_{\phi' \sim \psi'} \mathcal{V}(\mathrm{dom}(\psi'), \phi') \leq 0.5 \left( Q - \min_{\phi \sim \psi} (\mathcal{V}(\mathrm{dom}(\psi), \phi)) \right),$$

for any pair of history $\psi', \psi$ such that $\psi' \sim \psi$ and $p(\psi') \leq 0.5p(\psi)$. The relationship becomes obvious when we observe that Equation (3) can be written as $\mathcal{V}(S, \phi) = 1 - \sum_{\phi' \sim \phi(S)} p(\phi') = 1 - p(\psi)$, for all $\phi \sim \psi$ and choosing $Q = 1$. Hence,

$$LHS = 1 - \min_{\phi' \sim \psi'} (1 - p(\psi'))$$
$$= p(\psi')$$
$$\leq 0.5p(\psi)$$
$$= RHS$$

$\square$

**Proposition 2.** *Adaptive monotonicity and submodularity does not imply the MLRB and vice versa.*

*Proof.* We prove the proposition using two counter examples.

**Example 1.** *Consider an adaptive stochastic optimization problem with two items $X = \{a, b\}$ and two observations $O = \{0, 1\}$. There are four possible scenarios where both observations are possible at both locations and the prior over them is uniform. The function $f$ is defined such that $f(S, \phi) = |S \cap \{a\}|$ for all scenarios $\phi$. This example is trivially adaptive monotone submodular as $f$ does not depend on the scenario.*

*However, it is does not satisfy MLRB. Let history $\psi = \{\}$ and $\psi' = \{(b, 1)\}$. Hence, $p(\psi') \leq 0.5p(\psi)$. But $\hat{f}(dom(\psi), \psi) = \hat{f}(dom(\psi'), \psi') = 0$. Hence, there is no constant fraction $K > 1$ that fulfil Equation* (2).

**Example 2.** *Consider an adaptive stochastic optimization problem with two items $X = \{a, b\}$ and two observations $O = \{0, 1\}$, and maximum value $Q = 1$. The prior and function $f$ is defined in Table 1*

Table 1: $p$ and $f$ for Example 2

| $p(\phi)$ | $\phi$ | $\{\}$ | $\{a\}$ | $\{b\}$ | $\{a, b\}$ |
|---|---|---|---|---|---|
| 0.6 | (a,1) (b,0) | 0 | 1 | 0 | 1 |
| 0.4 | (a,0) (b,0) | 0 | 0.5 | 1 | 1 |

*This problem is pointwise monotone submodular. There are two pair of histories where $p(\psi') \leq 0.5p(\psi)$ and they are $\psi' = \{(a, 0)\}, \psi = \{\}$ and $\psi' = \{(a, 0), (b, 0)\}, \psi = \{(b, 0)\}$. For both pair histories, we can verify that they satisfy eq. (2) with upperbound $Q = 1$ and $K = 2$. Hence, this problem satisfies MLRB. On the other hand, $0.4 = \triangle(b|\{\}) < \triangle(b|\{(a, 0)\}) = 0.5$, it is not adaptive submodular.*

$\square$

**Proposition 3.** *The generalized version space reduction function $f_L$ satisfies MLB with constants $G = \max_{\phi, \phi'} L(\phi, \phi')$.*

*Proof.* The generalized version space reduction can be written as:

$$f_L(S, \phi) = \sum_{\phi'} p(\phi')L(\phi, \phi') - \sum_{\phi' \sim \phi(S)} p(\phi')L(\phi, \phi').$$

We also have

$$f_L(X, \phi) = \sum_{\phi'} p(\phi')L(\phi, \phi')$$

Let $G = \max_{\phi, \phi'} L(\phi, \phi')$. For any history $\psi$,

$$f_L(X, \phi) - f_L(\mathrm{dom}(\psi), \phi) = \sum_{\phi' \sim \phi(\mathrm{dom}(\psi))} p(\phi')L(\phi, \phi')$$

$$\leq \sum_{\phi' \sim \phi(\mathrm{dom}(\psi))} p(\phi') \cdot G$$

$$\leq G \cdot p(\psi)$$

and hence satisfies condition of MLB with constant $G = \max_{\phi, \phi'} L(\phi, \phi')$. $\qquad \square$

**Proposition 4.** *The Gibbs error function $f_{GE}$ is pointwise submodular and monotone. In addition, it satisfies condition MLRB with constants $Q = W(\mathcal{E}) = 1 - \sum_{i=1}^{m}(p(\mathcal{H}_i))^2$, the total weight of ambiguous pairs of hypotheses, and $K = 2$.*

*Proof.* First, we show $f_{GE}$ is pointwise submodular and monotone. For a fixed hypothesis $h \in H'$, the function $f_{GE}$ is monotone because it is the total weight of disambiguated pairs of hypotheses and the weight of a pair of hypotheses is nonnegative.

For a fixed hypothesis $h \in H'$, sets of location $A, B$, a location $y \notin B$, and $A \subseteq B$,

$$f_{GE}(A \cup \{y\}, h) - f_{GE}(A, h) = W(\cup_{x \in A} \mathcal{E}_x(h) \cup \mathcal{E}_y(h)) - W(\cup_{x \in A} \mathcal{E}_x(h))$$

$$= W(\mathcal{E}_y(h) \setminus \cup_{x \in A} \mathcal{E}_x(h))$$

$$\geq W(\mathcal{E}_y(h) \setminus \cup_{x \in B} \mathcal{E}_x(h))$$

$$= f_{GE}(B \cup \{y\}, h) - f_{GE}(B, h)$$

Hence $f_{GE}$ is submodular.

Now, we note that $Q - f_{GE}(\mathrm{dom}(\psi), h) = p(\psi)^2 - \sum_i p(\psi, \mathcal{H}_i)^2$. Given $p(\psi)$, the largest value for $\sum_i p(\psi, \mathcal{H}_i)^2$ occurs when there are only two equal valued probabilities $p(\psi, \mathcal{H}_1) = p(\psi, \mathcal{H}_2) = p(\psi)/2$ giving the value of $\sum_i p(\psi, \mathcal{H}_i)^2 = p(\psi)^2/2$ and $Q - f_{GE}(\mathrm{dom}(\psi), h) \geq p(\psi)^2/2$. When $p(\psi') \leq p(\psi)/2$, we have $p(\psi')^2 \leq p(\psi)^2/4$ and $Q - f_{GE}(\mathrm{dom}(\psi'), h) \leq p(\psi)^2/4$. Hence $Q - f_{GE}(\mathrm{dom}(\psi'), h) \leq p(\psi)^2/4 \leq (Q - f_{GE}(\mathrm{dom}(\psi), h))/2$ giving $K = 2$. $\qquad \square$

We now give the proofs for performance guarantees of RAC. For clarity, we refer to adaptive stochastic optimization problem on paths simply as adaptive stochastic optimization problem. Our proofs hold for both adaptive stochastic optimization problem on paths and on subsets unless we specifically specialize it to subsets at the end.

## 0.2 Approximate Submodular Orienteering

RAC uses submodular orienteering to choose the sequence of locations to visit to cover a submodular set function. Given a set of locations $X$, a metric $d$ that gives the distance between any pair of locations $x, x' \in X$, a starting location $r$, and a submodular function $f$ of the set of locations, the goal of submodular orienteering problem is to find a tour starting from $r$ that covers the function $f$. We use a three-steps SUBMODULARORIENTEER procedure that runs in polynomial time to approximate solution to a submodular orienteering problem. In the first step, we compute an approximation for distance metric $d$ with a tree [3]. Then we run a greedy approximation algorithm [1] for Polymatroid Steiner tree problem with the submodular function and approximation tree as input. Finally, we apply Christofides' metric TSP [2] to obtain an approximate solution.

**Lemma 1.** *Assuming the submodular function $f$ is integer-valued, the* SUBMODULARORIENTEER *procedure in RAC computes a $2\alpha$-approximation to the Submodular orienteering tour with $\alpha \in O((\log|X|)^{2+\epsilon} \log \nu)$ and $\nu = f(X)$.*

*Proof.* The greedy approximation in SUBMODULARORIENTEER computes an $\alpha$-approximation $T$ to the optimal polymatroid Steiner tree $T^*$, with $\alpha \in O((\log|X|)^{2+\epsilon}\log\nu)$, where $\nu$ is the required value [1]. The total edge-weight of an optimal polymatroid Steiner tree, $w(T^*)$, must be less than that of an optimal submodular orienteering tour, $W^*$, as we can remove any edge from a tour and turn it into a tree. Thus, $w(T) \leq \alpha\,w(T^*) \leq \alpha\,W^*$. Applying Christofides' metric TSP to the vertices of $T$ produces a tour $\tau$, which has weight $w(\tau) \leq 2w(T)$, using an argument similar to that in [2]. It then follows that $w(\tau) \leq 2\alpha W^*$. In other words, SUBMODULARORIENTEER obtains a $2\alpha$-approximation to the submodular orienteering tour. $\qquad\square$

## 0.3 Adaptive Stochastic Optimization on Paths

**Proposition 5.** *Let $f$ be a pointwise monotone submodular function. Then $g_\nu$ is pointwise monotone submodular and $g_\nu^*$ is monotone submodular. In addition $g_\nu^*(Z') \geq \nu$ if and only if $f$ is either covered or have value at least $\nu$ for all scenarios consistent with $\psi \cup Z'$.*

*Proof.* First note that the operations of adding a constant to a monotone submodular function, adding together one or more monotone submodular function and setting a ceiling to a monotone submodular function (taking the minimum of a function and a constant) all result in monotone submodular functions. Similarly, if $f_\nu(S, \phi)$ is monotone submodular for $X$, modifying it by setting $f_\nu(S, \phi) = f_\nu(X, \phi)$ if $S$ contains $x \in X$ preserves monotonicity and submodularity. To see this, note that $f_\nu(X, \phi)$ is the maximum value of the function and setting the function to its maximum later has less gain for a monotone function.

Note that $\min(\nu, g_\nu(Z', \phi))$, $g_\nu^*(Z') \geq \nu$ if and only if $g_\nu(Z', \phi) \geq \nu$ for all $\phi$. Finally, note that $g_\nu(Z', \phi) \geq \nu$ exactly when $Z'$ is inconsistent with $\phi$, or when it is consistent and $f(\mathrm{dom}(\psi \cup Z'), \phi)$ is covered, or when it is consistent and $f(\mathrm{dom}(\psi \cup Z'), \phi) \geq \nu$ as required.

$\qquad\square$

**Proposition 6.** *When $f$ satisfies minimal dependency, $g_\nu^m(Z') \geq \nu$ implies $g_\nu^*(Z') \geq \nu$.*

*Proof.* By definition, $g_\nu^m(Z') = g_\nu(Z', Z)$. As $f$ satisfies minimal dependency, $g_\nu$ also satisfies minimal dependency. Hence, if $g_\nu(Z', Z) \geq \nu$, we also have $g_\nu(Z', \phi) \geq \nu$ for all $\phi$, implying $g_\nu^*(Z') \geq \nu$

$\qquad\square$

We begin by analyzing a variant of adaptive stochastic optimization problem where the agent has to return to the starting location $r$ in the end. We assume that we can compute an optimal submodular orienteering solution, and then relax this assumption to use polynomial time approximation later. This subsection can be divided into three parts. First, we analyze RAC on problems satisfying the MLB condition (Lemma 2 to Lemma 7). Next, we complete the analysis for problems satisfying condition the MLRB condition (Lemma 8 to Lemma 10). Finally, we relax the assumptions of computing optimal submodular orienteering solution and of going back to the starting location. We derive the final approximation bounds for the non-rooted adaptive stochastic optimization problems satisfying the MLB condition and for those satisfying the MLRB condition (Lemma 11 to Theorem 1).

The main strategy of this analysis is to establish the post conditions upon termination of the adaptive plan in each recursive step. There are two components to prove in the post conditions; progress made in covering the function and distance traveled by the agent.

In the following (Lemmas 2 and 3), we show that each adaptive plan reduce likelihood of history by half except when it is the last recursive step where it completes the coverage.

**Lemma 2.** *Let $\tau$ be the solution to a submodular orienteering problem $g_\nu^*$ in GENERATETOUR1. Let $\psi$ be the history experienced by the agent after we call EXECUTEPLAN with tour $\tau$. Either $p(\psi) < 0.5$ or $g_\nu^*(\psi) = \nu$.*

*Proof.* During the execution of EXECUTEPLAN, if the agent receives an observation $o' \in \Omega_x$ at some location $x'$ on $\tau$, then the agent returns to $r$ immediately with history $\psi = ((x_1, o_1), \ldots, (x', o'))$. The probability of this history is $p(\psi) = \prod_{(x,o)\in\psi} p(o|x) \leq p(o'|x')$. From the definition of $\Omega_{x'}$, we have $p(\psi) \leq p(o'|x') < 0.5$.

Otherwise, the agent visits every location $x$ on $\tau$ and receives at every $x$ an observation $o_x^* \notin \Omega_x$ and has history $\psi = \psi^*(\tau)$, *i.e.* the agent always receive the most likely observation throughout the tour and $g_\nu^*(\psi) = \nu$. □

**Lemma 3.** *Let $\psi$ be the history after a recursive call of RAC. After each recursive call, either likelihood of history is reduced by half, $p(\psi) < 0.5$ or we have completely covered the function $f$.*

*Proof.* RAC calls EXECUTEPLAN with either $\tau_f$ or $\tau_{vs}$, which solves the submodular orienteering problem $g_Q^*$ and $\mathcal{V}_{0.5}^*$ respectively. If RAC uses $\tau_f$, Lemma 2 tells us that EXECUTEPLAN either reduces the likelihood of history by at least half or completely covers the function $g_Q^*$, which implies that we have completely covered the function $f$.

Otherwise, RAC uses $\tau_{vs}$ and reduces the version space (and equivalently $p(\psi)$) by at least a half.

Finally, we prove the lemma by combining the outcomes from using $\tau_f$ or $\tau_{vs}$. □

We want to bound the distance traveled in each recursive call by comparing the length of the submodular orienteering tour to a path in the optimal policy. This path always exist and is traversed with probability more than half by the optimal policy. Hence, we can bound the length of our tour by twice the expected cost of optimal policy.

**Lemma 4.** *Let $\pi^*$ be an optimal policy tree for a rooted adaptive stochastic optimization problem $\mathcal{I}$. There is a subpath $\sigma'$ of $\pi^*$ such that $\pi^*$ traverses $\sigma'$ with probability at least 0.5. Furthermore, one of the following conditions must hold: (1) the probability of most likely history on this path $p(\psi^*(\sigma')) \geq 0.5$ and $\psi^*(\sigma')$ covers $f$, or (2) $p(\psi^*(\sigma')) < 0.5$ and $p(\psi^*(\sigma'_{-1})) \geq 0.5$, where $\psi^*(\sigma'_{-1})$ is the most likely history without the final observation.*

*Proof.* We give the construction for such a subpath $\sigma'$. First, we extracts a path $\sigma$ from an optimal policy $\pi^*$ tree by following the most likely observation edge from the root. Let $\sigma = (r, x_1, x_2, \ldots, x_s, r)$ be a path in the optimal policy tree $\pi^*$ such that every edge following a node $x_i$ in the path is labeled with the most likely observation $o_{x_i}^* = \arg \max_{o \in O} p(o|x)$ up to the last node $x_s$ and then return to the root $r$. Thus, the history from traversing $\sigma$ is $\psi^*(\sigma)$.

Next, we need to ensure that $\pi^*$ traverses its subpath $\sigma'$ with probability at least 0.5. Let $p(\sigma_i|\pi^*)$ be the probability of reaching the node $x_i$ on the path $\sigma$ under the optimal policy $\pi^*$. It is equal the probability of traversing the path $\sigma$ and observing the most likely observation at every location in $\sigma$ up to $x_{i-1}$ and go on to $x_i$ (without making an observation at $x_i$) *i.e.*

$$p(\sigma_i|\pi^*) = p((r, (x_1, o_{x_1}^*), \ldots, (x_{i-1}, o_{x_{i-1}}^*), x_i))$$
$$= p(\psi^*(\sigma_{i-1}))$$

If $p(\sigma_s|\pi^*) < 0.5$, we truncate the path $\sigma_s$ from the end at a location $x_q$ such that $p(\sigma_q|\pi^*) > 0.5$. In other words, $\sigma_q$ is the longest subpath of $\sigma$ where $p(\sigma_q|\pi^*) > 0.5$. We set $\sigma' = (\sigma_q, r)$. That is, we return to the root $r$ after traversing $\sigma_q$. Otherwise $p(\sigma_s|\pi^*) \geq 0.5$, and we simply set $\sigma' = (\sigma_s, r) = \sigma$.

$\pi^*$ traverses $\sigma'$ with probability at least 0.5 by construction. If $\sigma' = \sigma$, it is a complete path along the most likely outcome branch from the root to the leaf of the optimal policy $\pi^*$. Thus, $f(\sigma', \phi) = f(X, \phi)$ for all scenarios $\phi \sim \psi^*(\sigma')$.

Otherwise, it is the truncated path $\sigma' = (\sigma_q, r)$. After receiving the most likely observation $o_{x_q}^*$ at $x_q$, we get $p((r, (x_1, o_{x_1}^*), \ldots, (x_q, o_{x_q}^*))) \leq 0.5$ because $\sigma_q$ is the longest subpath that is $p(\sigma_q|\pi^*) \geq 0.5$. Thus, $p(\psi^*(\sigma_q)) \leq 0.5$. □

**Lemma 5.** *Assuming we compute the optimal solution to the submodular orienteering problems, the agent travels at most $2C(\pi^*)$ for each recursive step of RAC.*

*Proof.* Using Lemma 4, we show that there is a subpath $\sigma'$ from the optimal policy $\pi^*$ that is a feasible solution to either the submodular orienteering problem $g_Q^*$ or $\mathcal{V}_{0.5}^*$.

Let $\sigma'$ a subpath from Lemma 4. If the first case of Lemma 4 is true , then $\sigma'$ is a feasible solution to the submodular orienteering problem $g_Q^*$. Otherwise the second case $p(\psi^*(\sigma')) < 0.5$

and $p(\psi^*(\sigma'_{-1})) \geq 0.5$, is true. Then $\sigma'$ is feasible solution to the problem of $\mathcal{V}_{0.5}^*$ because $\mathcal{V}_{0.5}(\sigma', \phi) = \min(0.5, 1 - p(\psi^*(\sigma'))) < 0.5$ for all scenario $\phi \in \Phi_{\sigma'}$.

Let $W_f^*$ and $W_{vs}^*$ be the total edge-weight of optimal submodular orienteering tour $\tau_f$ and $\tau_{vs}$ respectively. Let the total edge-weight of the tour used in each recursive step be $W^* = \min(W_f^*, W_{vs}^*)$. If it is the first case, then $W^* \leq W_f^* \leq W(\sigma')$. Otherwise, $W^* \leq W_{vs}^* \leq W(\sigma')$. As $\sigma'$ is traversed with probability at least 0.5,

$$C(\pi^*) \geq \sum_{\phi \sim \psi^*(\sigma)} p(\phi)w(\sigma')$$
$$\geq 0.5w(\sigma') \geq 0.5W^*$$
$$W^* \leq 2C(\pi^*),$$

where $w(\sigma')$ is the total edge-weight of tour $\sigma'$.

In EXECUTEPLAN, the agent travels on a path bounded by $W^*$. Hence, the agent travels at most $2C(\pi^*)$. $\qquad\square$

**Lemma 6.** *Suppose that $\pi^*$ is an optimal policy for a rooted adaptive stochastic optimization problem $\mathcal{I}$ with prior probability distribution $p$. Let $\{\Phi_1, \Phi_2, \ldots, \Phi_n\}$ be a partition of the scenarios $O^X$, and let $\pi_i^*$ be an optimal policy for the subproblem $\mathcal{I}_i$ with prior probability distribution $p_i$:*

$$p_i(\phi) = \begin{cases} p(\phi)/p(\Phi_i) & \text{if } \phi \in \Phi_i \\ 0 & \text{otherwise} \end{cases}$$

*where $p(\Phi_i) = \sum_{\phi \in \Phi_i} p(\phi)$ Then we have*

$$\sum_{i=1}^{n} p(\Phi_i)C(\pi_i^*) \leq C(\pi^*).$$

*Proof.* For each subproblem $\mathcal{I}_i$, we can construct a feasible policy $\pi_i$ for $\mathcal{I}_i$ from the optimal policy $\pi^*$ for $\mathcal{I}$. Consider the policy tree $\pi^*$. Every scenario $\phi$ must has a path $\sigma$ from root to the leaf in the optimal tree $\pi^*$ that covers the scenario because the optimal policy covers all scenarios. So we choose the policy tree $\pi_i$ as the subtree of $\pi^*$ that consists of all the paths that cover scenarios in $\Phi_i$. Clearly $\pi_i$ is feasible, as every scenario in $\Phi_i$ has a path in $\pi_i$ that covers it. Then,

$$\sum_{i=1}^{n} p(\Phi_i)C(\pi_i^*) \leq \sum_{i=1}^{n} p(\Phi_i)C(\pi_i)$$
$$\leq \sum_{i=1}^{n} p(\Phi_i) \sum_{\phi \in \Phi_i} \frac{p(\phi)}{p(\Phi_i)} \cdot C(\pi_i, \phi)$$
$$= \sum_{\phi \in \Phi_i} p(\phi)C(\pi^*, \phi) = C(\pi^*).$$

$\qquad\square$

For functions satisfying the MLB, the remaining objective value to cover is bounded by marginal likelihood of history multiplied by $G$. Every recursive call either reduces marginal likelihood of history by half or completely covers the function $f$ and thus bounding the remaining function to cover at the same time. The algorithms is repeated at most a logarithmic number of times and we can obtain an approximation bound.

**Lemma 7.** *Let $\pi$ denote the policy that RAC computes for a rooted adaptive stochastic optimization problem on paths. Let $\eta$ be any value such that $f(S, \phi) > f(X, \phi) - \eta$ implies $f(S, \phi) = f(X, \phi)$. Assume RAC computes an optimal submodular coverage tour in each step. If $f$ satisfies MLB, then for an instance of adaptive stochastic optimization optimizing $f$*

$$C(\pi) \leq 2\left(\log(G/\eta) + 1\right)C(\pi^*),$$

*where $C(\pi)$ is the expected cost of RAC.*

*Proof.* Let $\psi$ be the entire history experienced by the agent from the start of RAC. If a recursive call picks tour $\tau_f$, traverses the entire tour, and receive most likely observation throughout the tour, then $f(\text{dom}(\psi), \phi) = f(X, \phi)$ for all scenario $\phi \sim \psi$ and we have fully covered $f$. Otherwise, we repeat the recursive call until $f(X, \phi) - f(\text{dom}(\psi), \phi) < \eta$, for all $\phi \sim \psi$. The MLB condition gives us $f(X, \phi) - f(\text{dom}(\psi), \psi) \leq G \cdot p(\psi)$ for all $\phi \sim \psi$. Hence, we derive from Lemma 3 the number of recursive steps required for any scenario is at most $\log\left(\frac{G}{\eta}\right) + 1$.

We now complete the proof by induction on the number of recursive calls to RAId. For the base case of $k = 1$ call, $C(\pi) \leq 2C(\pi^*)$ by Lemma 5. Assume that $C(\pi) \leq 2(k-1)C(\pi^*)$ when there are at most $k - 1$ recursive calls. Now consider the induction step of $k$ calls. The first recursive call partitions the scearios into a collection of mutually exclusive subsets, $\Phi_1, \Phi_2, \ldots, \Phi_n$. Let $\mathcal{I}_i$ be the subproblem with scenario set $\Phi_i$ and optimal policy $\pi_i^*$, for $i = 1, 2, \ldots, n$. After the first recursive call, it takes at most $k - 1$ additional calls for each $\mathcal{I}_i$. In the first call, the agent incurs a cost at most $2C(\pi^*)$ by Lemma 5. For each $\mathcal{I}_i$, the agent incurs a cost at most $2(k-1)C(\pi_i^*)$ in the remaining $k - 1$ calls, by the induction hypothesis. Putting together this with Lemma 6, we conclude that the agent incurs a total cost of at most $2kC(\pi^*)$ when there are $k$ calls. $\qquad\square$

The MLRB condition (Equation (2)) tells us that we reduce the remaining function to cover by a fraction whenever the remaining version space is halved. Next, we show that the remaining function to cover is reduced by a fraction upon termination of each adaptive plan.

**Lemma 8.** *Let $\tau$ be the tour generated in a recursive and $\psi$ be the history after a recursive call of RAC. By the end of each recursive call, for each scenario $\phi \sim \psi$, $f(\text{dom}(\psi), \phi) \geq (1 - 1/K)Q$ unless $f(X, \phi) < (1 - 1/K)Q$. In that case, $f(\text{dom}(\psi), \phi) = f(X, \phi)$.*

*Proof.* The procedure EXECUTEPLAN is called with tour $\tau$ that is a solution to submodular orienteering problem $g^*_{(1-1/K)Q}$. From Lemma 2, if EXECUTEPLAN terminates with $p(\psi) \leq 0.5$, we know from MLRB (Equation (2)) that $f(\text{dom}(\psi), \phi) \geq (1 - 1/K)Q$ for all $\phi \sim \psi$. Otherwise, EXECUTEPLAN terminates with $g^*_{(1-1/K)Q}(\tau, \psi) = (1 - 1/K)Q$. In that case, from Proposition 4, $f(\text{dom}(\psi), \phi) \geq (1 - 1/K)Q$ or $f(X, \phi) < (1 - 1/K)Q$ and $f$ is already covered for $\phi$.

$\qquad\square$

**Lemma 9.** *Assuming we compute the optimal solution to the submodular orienteering problems, the agent travels at most $2C(\pi^*)$ for each recursive step of RAC.*

*Proof.* From Lemma 4 and MLRB, the subpath $\sigma'$ is feasible solution to the submodular orienteering problem of $g^*_{(1-1/K)Q}$. Let $W^*$ be the total edge-weight of the tour used in a recursive call of RAC. Then, $W^* \leq W(\sigma')$ because $W^*$ is the value of an optimal solution. Since $\sigma'$ is traversed with probability at least 0.5,

$$C(\pi^*) \geq \sum_{\phi \sim \psi^*(\sigma)} p(\phi) w(\sigma')$$
$$\geq 0.5 w(\sigma') \geq 0.5 W^*$$
$$W^* \leq 2C(\pi^*),$$

where $w(\sigma')$ is the total edge-weight of tour $\sigma'$.

In EXECUTEPLAN, the agent travels on a path bounded by $W^*$. Hence, the agent travels at most $2C(\pi^*)$. $\qquad\square$

**Lemma 10.** *Let $\pi$ denote the policy that RAC computes for a rooted adaptive stochastic optimization problem on paths. Let $\eta$ be any value such that $f(S, \phi) > f(X, \phi) - \eta$ implies $f(S, \phi) = f(X, \phi)$. Assume RAC computes an* optimal *submodular coverage tour in each step. If $f$ satisfies MLRB, then for an instance of adaptive stochastic optimization optimizing $f$*

$$C(\pi) \leq 2\left(\log_K(Q/\eta) + 1\right) C(\pi^*),$$

*where $C(\pi)$ is the expected cost of RAC, and $K > 1$ and $Q \geq \max_\phi f(X, \phi)$ are the constants that satisfy Equation (2).*

*Proof.* We need to repeat the recursive call until $f(X, \phi) - f(\text{dom}(\psi), \phi) \leq \eta$ for all $\phi \sim \psi$. From MLRB and Lemma 8, the number of recursive steps required for any scenario is at most $\log_K \left( \frac{Q}{\eta} \right) + 1$.

We now complete the proof by induction on the number of recursive calls to RAC. For the base case of $k = 1$ call, $C(\pi) \leq 2C(\pi^*)$ by Lemma 9. Assume that $C(\pi) \leq 2(k-1)C(\pi^*)$ when there are at most $k - 1$ recursive calls. Now consider the induction step of $k$ calls. The first recursive call partitions the scenarios into a collection of mutually exclusive subsets, $\Phi_1, \Phi_2, \ldots, \Phi_n$. Let $\mathcal{I}_i$ be the subproblem with scenario set $\Phi_i$ and optimal policy $\pi_i^*$, for $i = 1, 2, \ldots, n$. After the first recursive call, it takes at most $k - 1$ additional calls for each $\mathcal{I}_i$. In the first call, the agent incurs a cost at most $2C(\pi^*)$ by Lemma 9. For each $\mathcal{I}_i$, the agent incurs a cost at most $2(k-1)C(\pi_i^*)$ in the remaining $k - 1$ calls, by the induction hypothesis. Putting together this with Lemma 6, we conclude that the agent incurs a total cost of at most $2kC(\pi^*)$ when there are $k$ calls. Hence, we obtain our approximation bounds. $\qquad\square$

Now, we relax the optimal submodular orienteering assumption and replace it with our polynomial time approximation procedure.

**Lemma 11.** *An $\alpha$-approximation algorithm for rooted adaptive stochastic optimization problem on paths is a $2\alpha$-approximation algorithm for adaptive stochastic optimization.*

*Proof.* Let $C^*$ and $C_r^*$ be the expected cost of an optimal policy for an adaptive stochastic optimization problem and for a corresponding rooted adaptive stochastic optimization problem, respectively. As any policy for non-rooted problem can be turned into a policy for the root version by retracing the solution path back to the start location, we have $C_r^* \leq 2C^*$. An $\alpha$-approximation algorithm for rooted adaptive stochastic optimization computes a policy $\pi$ for $\mathcal{I}_r$ with expected cost $C_r(\pi) \leq \alpha C_r^*$. It then follows that $C_r(\pi) \leq \alpha C_r^* \leq 2\alpha C^*$ and this algorithm provides a $2\alpha$-approximation to the optimal solution of the non-rooted problem. $\qquad\square$

**Theorem 1.** *Assume that $f$ is an integer-valued pointwise submodular monotone function. If $f$ satisfies MLRB condition, then for any constant $\epsilon > 0$ and an instance of adaptive stochastic optimization problem on path optimizing $f$, RAC computes a policy $\pi$ in polynomial time such that*

$$C(\pi) = O((\log|X|)^{2+\epsilon} \log Q \log_K Q)C(\pi^*)),$$

*where $Q$ and $K > 1$ are constants that satisfies Equation (2).*

*Proof.* The distance traveled in each recursive step is at most $\alpha W^* \leq O(\alpha)C(\pi^*)$. From Lemma 1, the approximation factor for the submodular orienteering problem solved in RAC is $\alpha = O((\log|X|)^{2+\epsilon} \log Q)$. Putting this together with Lemma 10 with $\eta = 1$ since $f$ is integer-valued and Lemma 11, we get the desired approximation bound. The algorithm clearly runs in polynomial time. $\qquad\square$

**Theorem 2.** *Assume that prior probability distribution $p$ is represented as non-negative integers with $\sum_\phi p(\phi) = P$ and $f$ is an integer-valued pointwise submodular monotone function. If $f$ satisfies MLB, then for any constant $\epsilon > 0$ and an instance of adaptive stochastic optimization problem on path optimizing $f$, RAC computes a policy $\pi$ for in polynomial time such that*

$$C(\pi) = O((\log|X|)^{2+\epsilon}(\log P + \log Q) \log G)C(\pi^*),$$

*where $Q = \max_\phi f(X, \phi)$.*

*Proof.* Let $\alpha_1$ and $\alpha_2$ be the approximation factors when we compute the submodular orienteering tours $\tau_f$ and $\tau_{VS}$ respectively in one recursive call of RAC. Let the length of the tour chosen be $W$, Let the length of the tour chosen be $W$,

$$W = \min(\alpha_1 W_f^*, \alpha_2 W_{VS}^*)$$
$$\leq (\alpha_1 + \alpha_2)W^*$$
$$\leq 2(\alpha_f + \alpha_{VS})C(\pi^*)$$

The last inequality is due to Lemma 5. Hence, the distance traveled in each recursive step is at most $2(\alpha_f + \alpha_{VS})C(\pi^*)$. Lemma 1 tells us that $\alpha_1 \in O((\log|X|)^{2+\epsilon} \log Q)$ and $\alpha_2 \in O((\log|X|)^{2+\epsilon} \log P)$. Putting this together with Lemma 7 with $\eta = 1$ and Lemma 11, we get the desired approximation bound. The algorithm clearly runs in polynomial time. $\qquad\square$

## 0.4 Adaptive Stochastic Optimization on Sets

Adaptive stochastic minimum cost cover on sets (without path constraints) is a special case where the metric is a star graph where all elements are connected to a root node. In the special case of sets, the submodular orienteering problems that RAC solves become submodular set coverage problems. At the same time, the submodular orienteering procedure in RAC becomes a greedy selection policy where we always choose the element with highest value to cost ratio, $i.e. \max_{x \in X \setminus \text{dom}(\psi)} \frac{\triangle(x|\psi)}{c(x)}$.

**Lemma 12.** *Given a submodular set function $g : X \to \mathbb{R}$, let $\pi^G$ be the greedy selection policy. We have,*

$$C(\pi^G) \leq \left( 1 + \ln \frac{f(X) - f(\emptyset)}{f(X) - f(S^{T-1})} \right) C(\pi^*)$$

*where the subset $S^{T-1}$ is the set of elements selected before the last step of the greedy policy [4].*

Using Lemma 12, we can get tighter approximation bounds for stochastic sets functions and drop the integer representation assumption on the prior $p$.

**Theorem 3.** *Assume $f$ is an integer-valued pointwise submodular and monotone function. If $f$ satisfies MLRB condition, then for an instance of adaptive stochastic optimization problem on subsets optimizing $f$, RAC computes a policy $\pi$ in polynomial time such that*

$$C(\pi) = 4(\ln Q + 1)(\log_K Q + 1)C(\pi^*),$$

*where $Q$ and $K > 1$ are constants that satisfies Equation* (2).

*Proof.* The distance traveled in each recursive step is at most $\alpha W^* \leq 4\alpha C(\pi^*)$. From Lemma 12, the approximation factor for the submodular set cover problem solved in RAC is $\alpha = \log Q$. Putting this together with Lemma 10 with $\eta = 1$ and Lemma 11, we get the desired approximation bound. The algorithm clearly runs in polynomial time. $\square$

**Theorem 4.** *Assume $f$ is an integer-valued pointwise submodular and monotone function and $\delta = \min_\phi p(\phi)$. If $f$ satisfies MLB condition, then for an instance of adaptive stochastic optimization problem on subsets optimizing $f$, RAC computes a policy $\pi$ in polynomial time such that*

$$C(\pi) = 4(\ln 1/\delta + \ln Q + 2)(\log G + 1)C(\pi^*)),$$

*where $Q = \max_\phi f(X, \phi)$.*

*Proof.* Let $\alpha_1, \alpha_2$ be the approximation factors when we compute the submodular set cover $\tau_f$ and $\tau_{VS}$ respectively. Let the cost of the set of elements chosen be $W$,

$$\begin{aligned} W &= \min(\alpha_1 W_f^*, \alpha_2 W_{VS}^*) \\ &\leq (\alpha_1 + \alpha_2)W^* \\ &\leq 2(\alpha_f + \alpha_{VS})C(\pi^*) \end{aligned}$$

The last inequality is due to Lemma 5. Hence, the distance traveled in each recursive step is at most $4(\alpha_f + \alpha_{VS})C(\pi^*)$. From Lemma 12, the approximation factors for the submodular set cover problems are $\alpha_1 = \ln 1/\delta + 1$ and $\alpha_2 = \ln Q/\eta + 1$. Putting this together with Lemma 7 with $\eta = 1$ and Lemma 11, we get the desired approximation bound. The algorithm clearly runs in polynomial time. $\square$

# 1 Experiment Tasks

## 1.1 UAV Search and Rescue

In the UAV search and rescue task, an agent search for a victim in an area modeled as an $8 \times 8$ grid. Each grid cell has equal chance of containing the victim. The UAV can operate at two different altitudes. At the high altitude, it uses a noisy long-range sensor that determines whether the $3 \times 3$ grid around its current location contains the victim. The sensor has a 0.03 chance of reporting the opposite reading. At the low altitude, the UAV uses an accurate short-range sensor that determines whether the current grid cell contains the victim.

The movement cost between two grid cells on the same altitude is the Manhattan distance between them multiplied by 4 at the high altitude and multiplied by 1 at the low altitude. The cost to move between high and low altitudes is 10. The victim is deemed to be safe if we know that he is in the safe zone (marked grey in Figure 1), otherwise we need to know the exact location of the victim. The equivalence classes task are the safe zone and every location outside of it.

## 1.2 Grasping a Cup

In a noisy variant of the grasping task, a robot arm needs to identify the cup with a handle among two cups on the table and lift it up by grasping the handle (Figure 2). The cups' positions are detected using an external camera on the left side of the table but it is uncertain which cup has the handle and where the handle is due to occlusion. Each hypothesis is a tuple of two parameters: $\kappa$ indicates which cup has a handle, and $\theta$ determines the handle location. The hypotheses where the handle faces away from the external camera have higher prior probabilities.

The robot arm has a single-beam laser range finder mounted at its the wrist to detect distance to the nearest object in the direction it is facing. It has a $0.85$ chance of detecting the correct discretized value $x$, $0.05$ chance of $+1$ or $-1$ error each, and $0.025$ chance of $+2$ or $-2$ errors each. We sample seven wrist positions $x_1, x_2, \ldots, x_7$ around the cups (Figure 2). At each position, the robot can pan the range finder in the plane parallel to the tabletop incurring a fixed cost of 4. Moving the wrist from one position to another incurs a higher cost of the distance between the two positions multiplied by 15. The robot arm starts at wrist position $x_1$ on the left side of the table.

The robot gripper is fairly robust to estimation error of the cup handle's orientation. For each cup, we partition the cup handle orientation into regions of 20 degrees each. We only need to know the region that contains cup handle. The equivalence classes here are the regions. However, it is not always possible to identify the true region due to observation noise. We can still reduce to ECD problem by associating each observation vector to its most likely equivalence class.