[Reviews · NeurIPS 2015]

Submitted by Assigned_Reviewer_1

*Summary*

This paper studies the adaptive stochastic optimization on paths, by imposing some constraints (namely the MLRB and the MLB conditions) on the relationship between the probability of a realization and its utility. While the notion of *adaptive submodularity* states that in expectation, no item will "surprise" us given a longer history, the MLRB / MLB conditions in this paper roughly characterizes the fact that such "surprise" only happens with small probability (i.e., realization with small probability has large utility). By leveraging such constraints, the authors propose a novel recursive algorithm with near-optimal guarantee on the expected cost needed to reach a certain coverage. Under the MLB conditions, the paper recovers the classical approximation result on bayesian pool-based active learning, up to a constant factor.

*Quality*

The paper is nicely written and well structured. Regarding the experimental results: by Theorem 4, it would be very interesting to see how the proposed RAC algorithm competes with existing algorithms *empirically* on the pool-based active learning problem (e.g., [3][5][7], or [6] under the bounded noise setting), given the fact that the version space reduction function and the Gibbs error function both satisfy MLRB condition, pointwise submodularity condition, and adaptive submodularity condition. Having these results will certainly make the paper stronger.

*Clarity*

Most of the paper is clear. It would be helpful if the authors elaborate more on the intuitive explanations of the two conditions: why are those two conditions more natural than the adaptive submodularity condition for adaptive optimization problems on paths?

*Originality and Significance*

To the best of my knowledge, the technical contribution of this contribution is novel. It provides interesting insights to developing tractable algorithms for the adaptive stochastic optimization problem.
Summary: The paper proposes an interesting solution to the adaptive stochastic optimization problem on paths (and sets as a special case). The authors identify a class of problems that can be near-optimally approximated using their solution, and provides strong empirical evidence. The presentation is in general clear and easy to read.

Submitted by Assigned_Reviewer_2

This paper gives an iterative procedure to solve the Adaptive Stochastic Optimization problem. The problem defined by an objective function f, and the paper develops conditions on the function that make the problem amenable to their solution approach (recursive adaptive convergence).

Quality: The ASO problem is described as a generalization of POMDP and conditions on the solution technique are given in simulation. The choice of baseline algorithms makes sense, but the paper would be much stronger if it the approach was compared more directly to similar approaches, e.g. popular POMDP solutions. Since the paper is rather technical giving more practical/concrete examples of the propositions would be helpful.

Clarity: The layout of the paper, language and flow are excellent, especially given the technical nature of the material.

Connecting the flow to specific problems is difficult. Having a running example, might make the paper easier to follow.

Originality:

This paper seems to capitalize on theoretical developments in sub-modular function optimization and applying these results to the ASO problem. The original part comes from the particular application.

Significance: Given the generality of the setting, the results could have a significant impact.

However, the current presentation does not make a strong case for this potential. Comparing the numerical results to alternative solution techniques or more directly arguing for the applicably of the MLRB and MLB conditions would make the case of impact stronger.

Summary: This paper gives a tractable

solution algorithm and sufficient conditions for when it's applicable for a general (and generally intractable) planning problem (ASO).

Submitted by Assigned_Reviewer_3

This paper is generally clear, with a few exceptions as noted below.

The paper extends current work in a potentially useful way, although the experiments used to illustrate the method are fairly small and simple.

The theoretical work relates this work to prior work (prop 2), demonstrates applicability to a new domain (prop 1 and 3), and analyzes the algorithm.

The usefulness of the algorithm analysis is questionable (e.g. Thm 1 and 2).

The resulting bound grows with the square of the log of the number of actions (where this is actually the number of possible locations on the path, as each action is "visit location x") as compared to the optimal cost.

This does not seem especially tight for application with any reasonable-sized problem.

A couple of clarity issues: line 110, should this inequality be reversed as A is a subset of B? line 125, what are the conditions for termination of a policy and how is that determined in practice? line between 151 and 152, I think some mistakes in notation line 216, what are alignment assumptions? experiments: the labels on the plots don't match those in the text.

The captions should be more informative and not repeat information already on the plots (Gibbs error)
Summary: This paper introduces an extension to submodular optimization that extends the method to apply to more than sets.

The contributions include the development and analysis of two conditions that allow this, as well as introducing an algorithm to solve problems meeting these conditions.

These contributions appear novel, well justified, and of interest to the community.

Author Feedback
Author rebuttal: Assigned Reviewer 1:

*Quality*

The paper is nicely written and well structured. Regarding the experimental results: by Theorem 4, it would be very interesting to see how the proposed RAC algorithm competes with existing algorithms *empirically* on the pool-based active learning problem (e.g., [3][5][7], or [6] under the bounded noise setting), given the fact that the version space reduction function and the Gibbs error function both satisfy MLRB condition, pointwise submodularity condition, and adaptive submodularity condition. Having these results will certainly make the paper stronger.

A >>>
We thank reviewer for feedback. We intend to compare with algorithms on active learning problems in an extended version of this paper.

*Clarity*

Most of the paper is clear. It would be helpful if the authors elaborate more on the intuitive explanations of the two conditions: why are those two conditions more natural than the adaptive submodularity condition for adaptive optimization problems on paths?

A >>>
We do not know of any method to exploit adaptive submodularity condition for adaptive optimization problems on PATHS. The two conditions seem natural to us because we can plan a path assuming the maximum likelihood observation and exploit the condition if we don't encounter the ML on the path. We will elaborate on this issue in the revision.

Assigned reviewer 2:

Quality: The ASO problem is described as a generalization of POMDP and conditions on the solution technique are given in simulation. The choice of baseline algorithms makes sense, but the paper would be much stronger if it the approach was compared more directly to similar approaches, e.g. popular POMDP solutions. Since the paper is rather technical giving more practical/concrete examples of the propositions would be helpful.

A >>>
We thank reviewer for feedback. We intend to compare with POMDP algorithms in an extended version of this paper.

Significance: Given the generality of the setting, the results could have a significant impact. However, the current presentation does not make a strong case for this potential. Comparing the numerical results to alternative solution techniques or more directly arguing for the applicably of the MLRB and MLB conditions would make the case of impact stronger.

A >>>
Thank you for the suggestion. We will try to think about meaningful alternatives for comparison.

Assigned reviewer 3:

This paper is generally clear, with a few exceptions as noted below. The paper extends current work in a potentially useful way, although the experiments used to illustrate the method are fairly small and simple.

The theoretical work relates this work to prior work (prop 2), demonstrates applicability to a new domain (prop 1 and 3), and analyzes the algorithm. The usefulness of the algorithm analysis is questionable (e.g. Thm 1 and 2). The resulting bound grows with the square of the log of the number of actions (where this is actually the number of possible locations on the path, as each action is "visit location x") as compared to the optimal cost. This does not seem especially tight for application with any reasonable-sized problem.

A >>>
We agree that the bounds are not as tight as we like due to log |X|^{2+e} factor. Unfortunately, this is derived from a state-of-the-art polynomial time polymatroid path planning algorithm. We speculate that the algorithm will be good for most practical cases. Otherwise, we can replace the polynomial time path planning with general search algorithms

A couple of clarity issues:
line 110, should this inequality be reversed as A is a subset of B?

A >>>
Yes, we will correct that in revision.

line 125, what are the conditions for termination of a policy and how is that determined in practice?

A >>>
We terminate when the worst case value of the function given a history is maximum. It will be easier to evaluate termination if the function satisfies minimal dependency condition (prop. 5).

line between 151 and 152, I think some mistakes in notation
A >>>
I think it is correct. However, we intend to revise the notations to be more consistent with Golovin and Krause paper for clarity.

line 216, what are alignment assumptions?

A >>>
We will revise this sentence and bring the short discussion of alignment forward.

experiments: the labels on the plots don't match those in the text. The captions should be more informative and not repeat information already on the plots (Gibbs error)

A >>>
We will correct the mistake in the label. We agree the captions can be simplified.

Assigned reviewer 4:

Thank you for your encouragement.

Assigned reviewer 6:

We thank reviewer for feedback. We will clarify the notations especially for the prior and posterior distributions in the revision.